# Pyridoxamine Alleviates Cardiac Fibrosis and Oxidative Stress in Western Diet-Induced Prediabetic Rats

**DOI:** 10.3390/ijms25158508

**Published:** 2024-08-04

**Authors:** Sarah D’Haese, Lisa Claes, Eva Jaeken, Dorien Deluyker, Lize Evens, Ellen Heeren, Sibren Haesen, Lotte Vastmans, Ivo Lambrichts, Kristiaan Wouters, Casper G. Schalkwijk, Dominique Hansen, BO Eijnde, Virginie Bito

**Affiliations:** 1UHasselt, Cardio & Organ Systems (COST), Biomedical Research Institute, Agoralaan, 3590 Diepenbeek, Belgium; sarah.dhaese@uhasselt.be (S.D.); dorien.deluyker@uhasselt.be (D.D.); ellen.heeren@uhasselt.be (E.H.); sibren.haesen@uhasselt.be (S.H.); lotte.vastmans@uhasselt.be (L.V.); ivo.lambrichts@uhasselt.be (I.L.); 2Department of Internal Medicine, CARIM School for Cardiovascular Diseases, Maastricht University Medical Centre, Universiteitssingel 50, 6229 ER Maastricht, The Netherlands; kristiaan.wouters@maastrichtuniversity.nl (K.W.); c.schalkwijk@maastrichtuniversity.nl (C.G.S.); 3UHasselt, Faculty of Rehabilitation Sciences, REVAL Rehabilitation Research Centre, Agoralaan, 3590 Diepenbeek, Belgium; dominique.hansen@uhasselt.be; 4Department of Cardiology, Heart Centre Hasselt, Jessa Hospital, Stadsomvaart 11, 3500 Hasselt, Belgium; 5SMRc-Sports Medicine Research Center, BIOMED-Biomedical Research Institute, Faculty of Medicine & Life Sciences, Hasselt University, 3590 Diepenbeek, Belgium; bert.opteijnde@uhasselt.be; 6Division of Sport Science, Stellenbosch University, Stellenbosch 7602, South Africa

**Keywords:** pyridoxamine, type 2 diabetes, Western diet, adverse cardiac remodeling, prevention, cardioprotection

## Abstract

Individuals with type 2 diabetes mellitus (T2DM) are at an increased risk for heart failure, yet preventive cardiac care is suboptimal in this population. Pyridoxamine (PM), a vitamin B_6_ analog, has been shown to exert protective effects in metabolic and cardiovascular diseases. In this study, we aimed to investigate whether PM limits adverse cardiac remodeling and dysfunction in rats who develop T2DM. Male rats received a standard chow diet or Western diet (WD) for 18 weeks to induce prediabetes. One WD group received additional PM (1 g/L) via drinking water. Glucose tolerance was assessed with a 1 h oral glucose tolerance test. Cardiac function was evaluated using echocardiography and hemodynamic measurements. Histology on left ventricular (LV) tissue was performed. Treatment with PM prevented the increase in fasting plasma glucose levels compared to WD-fed rats (*p* < 0.05). LV cardiac dilation tended to be prevented using PM supplementation. In LV tissue, PM limited an increase in interstitial collagen deposition (*p* < 0.05) seen in WD-fed rats. PM tended to decrease 3-nitrotyrosine and significantly lowered 4-hydroxynonenal content compared to WD-fed rats. We conclude that PM alleviates interstitial fibrosis and oxidative stress in the hearts of WD-induced prediabetic rats.

## 1. Introduction

More than 536 million people are currently affected by diabetes mellitus and its global incidence keeps rising at alarming rates [1]. Individuals with type 2 diabetes mellitus (T2DM) are at a higher risk for cardiovascular diseases (CVD) [2]. Diabetic cardiomyopathy (DCM) includes all diabetes-induced structural and functional changes in the myocardium in the absence of other cardiac risk factors, such as hypertension, coronary artery disease, and valvular disorders [3]. The pathogenesis of DCM is multifaceted and complex [4]. Metabolic changes during the development of T2DM (i.e., hyperglycemia, hyperinsulinemia, and lipotoxicity) induce microvascular endothelial dysfunction and a state of oxidative stress and inflammation in the myocardium [5]. Altogether, these pathological mechanisms trigger increased collagen deposition by fibroblasts and stimulate cardiomyocyte hypertrophy, progressively inducing a stiffer heart [5]. In particular, the formation and deposition of advanced glycation end products (AGEs) are thought to contribute to excessive oxidative stress and fibrosis in the heart [6] and large blood vessels [7]. Initially, T2DM patients can therefore present with diastolic dysfunction or heart failure with preserved ejection fraction (HFpEF) [8]. This so-called restrictive phenotype of DCM is characterized by a normal left ventricular (LV) ejection fraction (EF), increased LV wall thickness, and elevated LV-filling pressures. High concentrations of reactive oxygen species (ROS) and AGEs, and the presence of hypoxia caused by significant microvascular dysfunction, might lead to further adverse cardiac remodeling with cardiomyocyte damage and apoptosis [5]. Although less common, these pathological processes might cause systolic dysfunction or heart failure with reduced ejection fraction (HFrEF), also known as the dilated phenotype of DCM, in patients with T2DM [8]. Accordingly, this phenotype of DCM is characterized by reduced EF, LV dilation, and contractile dysfunction. Whereas some researchers state that the two DCM phenotypes progress chronologically over time, others believe that DCM encompasses two predestined, distinct phenotypes [5].

The current management of individuals with T2DM and HFrEF primarily consists of multiple drugs as well as device therapies to improve cardiac health [8]. For diabetes-related HFpEF, sodium-glucose cotransporter 2 inhibitors recently have been shown to be effective in lowering the risk of cardiovascular death or hospitalization for heart failure substantially [9,10]. Besides tackling cardiac complications, the prescription of antidiabetic drugs is also required [11]. Although heart failure in T2DM is currently managed by a combination of multiple pharmacological strategies, no specific compound is available to simultaneously prevent metabolic and cardiac adverse changes in T2DM. Pyridoxamine (PM), a vitamin B_6_ analog, has been shown to exert protective effects in various metabolic and cardiovascular diseases [12,13,14,15,16,17]. As such, preclinical studies have demonstrated that PM can prevent atherosclerosis [17], arterial stiffness [13,16], chemotherapy-induced cardiomyopathy [15], and diabetes- or obesity-related nephropathy [12,14]. In general, the positive effects of PM can be attributed to its capability to inhibit inflammation, trap ROS, and quench AGEs, the latter by scavenging methylglyoxal, which is a major precursor in AGEs formation [18,19]. To date, it is unknown whether PM can prevent cardiac dysfunction in the setting of T2DM. In this study, we aimed to investigate whether and how PM limits the onset of adverse cardiac remodeling and cardiac dysfunction in a prediabetic rat model. We hypothesize that PM offers cardioprotection in diet-induced prediabetic rats, mediated by decreased LV cardiac fibrosis and oxidative stress.

## 2. Results

### 2.1. Pyridoxamine Improves Fasting Blood Glucose Levels in Western Diet-Fed Rats

The Western diet (WD) regime led to a significant body weight gain compared to a standard chow diet (CD) from week 9 onwards (Figure 1A). PM did not affect body weight gain. Tibia length remained comparable in all groups (Appendix A). In line, WD with or without PM supplementation induced a significantly elevated heart and liver weight compared to CD animals (Figure 1B,C and Appendix A). In addition, the WD group tended to display heavier lungs (Figure 1D and Appendix A). No differences were observed in kidney weight between the groups (Appendix A). Altogether, PM did not prevent the WD-induced increase in body, heart, and liver weight in rats.

WD-fed rats showed at 6, 12, and 18 weeks a significantly lower glucose tolerance as compared to CD rats (Figure 2A). Although not significant, the glucose tolerance seemed to be improved with PM administration in WD-fed rats after 12 weeks and 18 weeks.

In addition, animals undergoing WD had significantly increased fasting plasma glucose levels compared to CD rats (Figure 2B). Interestingly, PM prevented this increase in the fasting plasma glucose levels seen in WD rats. Regarding fasting plasma insulin levels, no significant changes were observed between the groups. The homeostasis model assessment of insulin resistance (HOMA-IR) was comparable between the groups after 18 weeks (Appendix A). Plasma glucose levels were significantly increased in the group with WD and drinking water supplemented with PM (WD + PM) and, although not significant, seemed to be increased in the WD group compared to the CD group 60 min after oral glucose administration (Figure 2C).

### 2.2. Pyridoxamine Tends to Prevent Western Diet-Induced LV Dilation

Echocardiography was performed after 18 weeks to assess the in vivo LV cardiac function (Table 1). The WD group showed a significantly increased systolic area (A_s_) and end-systolic volume (ESV) which tended to be attenuated using PM administration. In addition, rats receiving a WD tended to have a reduced EF compared to CD animals, but this was not improved with PM treatment. Longitudinal and radial fractional shortening (FS), internal diameters in the diastole and systole (ID_d_ and ID_s_), stroke volume (SV) index, and cardiac index were similar between all groups (Appendix A). Echocardiographic parameters for LV hypertrophy, namely left ventricular posterior wall thickness in the diastole and systole (PWT_d_ and PWT_s_), anterior wall thickness in the diastole and systole (AWT_d_ and AWT_s_), interventricular septum thickness in the diastole and systole (IVS_d_ and IVS_s_), the ratio of peak mitral flow velocity in early versus late filling (E/A), and the ratio of peak mitral flow velocity versus peak mitral annular velocity (E/E’) remained comparable in all groups (Appendix A). Hemodynamic parameters regarding LV filling (i.e., end-diastolic pressure (EDP)), isovolumetric relaxation (i.e., time constant for isovolumetric relaxation (Tau)), and hypertension (i.e., end-systolic pressure (ESP)) were similar for all groups (Appendix A).

### 2.3. Pyridoxamine Limits LV Fibrosis and Oxidative Stress in Western Diet-Fed Rats

LV tissue sections were stained with Sirius Red/Fast Green staining to visualize and semi-quantify interstitial collagen deposition (Figure 3A). LV interstitial fibrosis was significantly increased in the WD group compared to the CD group (Figure 3B). Accordingly, PM administration significantly prevented the increase in collagen deposition seen in the WD group. In addition, we investigated the expression of genes involved in cardiac fibrosis with RT-qPCR. The gene expression of collagen type I and III was similar in the LV of WD animals and CD rats (Figure 3C and Appendix A). PM significantly downregulated the expression of type I collagen compared to the WD group (Figure 3C). Furthermore, LV tissue sections were stained with hematoxylin and eosin (H&E) to evaluate the cross-sectional cardiomyocyte area (CSA). Representative images of the H&E staining are shown in Appendix A. The CSA was significantly increased in the WD + PM group compared to the CD group (Appendix A).

To further unravel the potential underlying beneficial mechanisms of PM, we stained LV tissue for the oxidative stress markers 3-nitrotyrosine and 4-hydroxynonenal (4-HNE) and evaluated the expression of genes involved in redox homeostasis. Representative pictures of LV sections stained for 3-nitrotyrosine are shown in Figure 4A,B. 3-nitrotyrosine deposition in the LV of the WD + PM group was significantly decreased compared to the CD group and tended to be lower compared to the WD group (Figure 4C). Furthermore, PM significantly decreased 4-HNE deposition in the LV of the WD + PM group compared to the CD and WD group (Figure 4D). In addition, PM significantly downregulated the gene expression of nicotinamide adenine dinucleotide phosphate oxidase 4 (NOX4) compared to the WD group (Figure 4E). The gene expression of antioxidants glutathione peroxidase 1 (GPx1) and superoxide dismutase (SOD2) in LV tissue remained similar in all groups (Appendix A).

Besides investigating LV tissue, we also stained aortic sections with Sirius Red/Fast Green and 3-nitrotyrosine to unravel the effect of PM on vascular remodeling and oxidative stress, respectively (Appendix A). The deposition of collagen (Appendix A) and 3-nitrotyrosine (Appendix A) in the aorta was comparable for all groups.

Lastly, to identify whether PM affects AGEs in the myocardium and plasma of WD-fed rats, the total AGEs content was semi-quantified from stained LV sections and measured in plasma samples via ELISA, respectively. The total AGEs content in the myocardium and plasma remained comparable between the three groups (Figure 5A–C). Furthermore, we evaluated with RT-qPCR whether LV gene expression of the receptor for advanced glycation end products (RAGE) in WD-fed rats was affected by PM administration (Figure 5D). The gene expression of RAGE in LV tissue did not differ between the groups.

## 3. Discussion

In this study, we are the first to show the cardioprotective effects of PM in a WD-induced prediabetic rat model. We have demonstrated that PM alleviates interstitial fibrosis and oxidative stress in the heart of WD-induced prediabetic rats.

### 3.1. A Translatable Rat Model of Diet-Induced Diabetic Cardiomyopathy

To study the underlying mechanisms in DCM and test the effectiveness of pharmacological compounds against the disease, various animal models are available to date [20,21]. Malik et al. showed that long-term, high intake levels of sugar-sweetened beverages are positively associated with the development of T2DM [22] and CVD mortality [23]. Furthermore, Velagic et al. demonstrated that adverse cardiac remodeling and cardiac dysfunction were more apparent in rats with T2DM induced by a high-sucrose high-fat diet (HFD) when compared to those on a moderate-sucrose HFD, suggesting dietary sucrose as an important mediator in DCM [24]. In this regard, we developed a rat model for DCM induced by high sucrose or a so-called WD to align the clinical situation [25]. We previously demonstrated that rats fed a WD for 18 weeks display metabolic changes including glucose intolerance, insulin resistance, and increased plasma triglyceride levels, thereby indicating the presence of (pre)diabetes. In addition, WD-fed rats developed the restrictive phenotype of DCM, as they have LV hypertrophy, interstitial fibrosis, and diastolic dysfunction. Prolonged feeding, namely for a total duration of 30 weeks, led to the dilated phenotype of DCM characterized by LV dilation, interstitial fibrosis, and systolic dysfunction [26]. In the current study, we induced the dilated or HFrEF phenotype of DCM as indicated by hyperglycemia, increased LV cardiac volumes, cardiac interstitial fibrosis, and reduced EF by feeding rats a WD for 18 weeks. Based on the findings in the current study, we acknowledge that oxidative stress (i.e., 3-nitrotyrosine and 4-HNE deposition) and AGEs do not seem to fully contribute to the WD-induced cardiac fibrosis. Further investigation into systemic lipid metabolism and systemic and cardiac inflammation in WD-fed rats, which potentially underly the increased cardiac collagen deposition, is warranted [4,26].

### 3.2. PM as a Potential Cardioprotective Strategy in T2DM Development

The relationship between diabetes and CVD is complex and bidirectional [5]. Metabolically, we have demonstrated that PM administration halted an increase in fasting glucose levels and, although not significant, could potentially prevent glucose intolerance in WD-fed rats, validating findings of others in HFD-induced diabetic and obese rodents [27,28,29]. These studies also showed that HFD consumption was associated with insulin resistance, body weight gain, and adiposity, which were all reduced using PM administration. Although we did see a decrease in fasting blood glucose levels, we could not detect any effects of PM on insulin sensitivity or body weight in this study. It should be noted, however, that the HFD-fed rodents are extreme models compared to our relatively mild T2DM phenotype induced by high sucrose in rats. In addition, the applied PM concentration and duration also differ between these preclinical studies, making a comparison of the metabolic effectivity of PM complex. In line with our findings, Chiazza et al. demonstrated that 1 g/L PM for 9 weeks improved hyperglycemia but did not influence insulin levels or body weight in mice fed a high-fructose and high-fat diet [30].

PM supplementation has been shown to prevent the development of diet-related kidney dysfunction [30,31], retinopathy [32], and vascular calcification [33], which are important complications of T2DM. Here, we are the first to demonstrate that PM partially prevents systolic dysfunction, as it tends to halt the development of LV-dilated cardiomyopathy but does not influence the reduced EF seen in WD-fed rats. Furthermore, we did not identify changes in echocardiographic parameters for LV cardiac hypertrophy (i.e., unchanged wall thicknesses) between the groups. At the cellular level, however, we showed that the CSA was significantly increased in the WD + PM group and, although not significant, it seemed to be increased in the WD group compared to the CD group. This discrepancy between cellular and echocardiographic measurements of hypertrophy might be explained by the time-dependent progression of the restrictive DCM phenotype into the dilated DCM phenotype [5,34]. As such, 18 weeks of WD might induce the enlargement of cardiomyocytes, as observed by an increased CSA, and simultaneously stimulate the lengthening of cardiomyocytes, leading to LV chamber dilation. Furthermore, we did not observe changes in diastolic function (i.e., EDP, E/A, E/E’) between the groups. These results align with those of Maessen et al., who failed to observe any effects of PM on EF, ESP, or EDP in HFD-fed mice [28]. Nevertheless, Haesen et al. demonstrated that PM attenuated chemotherapy-induced LV-dilated cardiomyopathy (e.g., reduced LV volumes) and systolic dysfunction (e.g., improved EF), as it prevents an increase in interstitial LV fibrosis, inflammation, and mitochondrial damage [15]. In addition, Deluyker et al. showed that pre-treatment with PM ameliorated diastolic function in rats with myocardial infarction, which was related to reduced peri-infarct collagen type I and III content [35]. Here, we show similar effects in WD-induced prediabetic rats as PM limits an increase in LV interstitial fibrosis, potentially through the reduced gene expression of collagen type I. It is likely that the beneficial effects of PM on cardiac fibrosis are due to improved tissue turnover. As such, PM has been shown to reduce hepatic fibrosis by lowering protein concentrations of matrix metalloproteinases 2 and 9, which are matrix-degrading enzymes, in a liver injury rat model [36]. Nevertheless, the effects of PM on the cardiac expression of genes involved in fibrosis and extracellular matrix remodeling (e.g., matrix metalloproteinases, Smad proteins, and transforming growth factor-β) remain to be confirmed in our animal model.

In addition, PM has been shown to improve diastolic dysfunction and limit carbonylation on cardiac ryanodine receptors and sarco(endo)plasmic reticulum Ca^2+^-ATPase in chemically induced type 1 diabetic rats [37,38]. This indicates that PM might limit cardiac dysfunction by impeding the protein adduct formation, potentially caused by ROS, reactive nitrogen species, and/or reactive carbonyl species (RCS), on important players of the excitation–contraction coupling. In this study, we have demonstrated that PM reduces and tends to lower 3-nitrotyrosine content in the LV compared to CD-fed rats and WD-fed rats, respectively. 3-nitrotyrosine is a protein modification of peroxynitrite, which in turn is formed when superoxide anions and nitric oxide react in the cardiovascular system [39]. We also confirmed the potential of PM to lower cardiac superoxide formation. As such, the WD + PM group had a decreased LV gene expression of NOX4, a mitochondrial enzyme that produces hydrogen peroxidase and superoxide radicals, compared to the WD group. In addition, we have shown that PM alleviates lipid peroxidation in the LV, as it decreases 4-HNE deposition compared to CD- and WD-fed rats. 4-HNE is a reactive lipid aldehyde generated via ROS-induced lipid peroxidation and is considered an important indicator of oxidative stress in T2DM with heart failure [40,41]. In line with our findings, PM was shown to lower hydrogen peroxide and lipid peroxidation in the serum of HFD-fed mice [27]. In vitro, pre-treatment with PM also prevented an increase in hydrogen peroxide and lipid peroxidation, whereas it stimulated the activity of endogenous antioxidants in high glucose-incubated renal cells [42]. Altogether, our findings on multiple markers of oxidative stress confirm that PM potentially acts as an antioxidant agent in the LV of WD-fed rats.

Regarding glycation, PM has been shown to fully or partially lower RCS, including methylglyoxal or AGEs content in the adipose [28,29] and renal [27] tissue of HFD-fed rodents. In our study, PM did not affect the overall LV myocardial AGEs content. As we measured total LV AGEs content semi-quantitatively, future research should identify whether and which specific RCS or AGEs in cardiac tissue are affected using PM via ultra-performance liquid chromatography tandem mass spectrometry [43].

Besides improving cardiac function, PM is known to also affect macrovascular structure and function [13,16,28,44,45,46]. As such, Huang et al. showed attenuated atherosclerosis with PM treatment in the aorta of genetically modified mice fed a high-sucrose, moderate-fat diet [44]. The authors report that the beneficial effect of PM on atherosclerotic plaques is likely attributable to the prevention of cellular oxidative damage. Furthermore, aortic collagen deposition has also been found to be attenuated using PM administration in chemically induced type 1 diabetic mice [13]. Functionally, PM was shown to partially prevent mild vascular dysfunction in isolated aortas from HFD-fed mice [28]. Here, neither WD nor PM induced aortic structural changes, as collagen and 3-nitrotyrosine deposition remained comparable between the groups. It might be that a longer study period is required to observe structural aortic adaptations and the protective effects of PM in animal models. As a consequence, the T2DM phenotype of our rat model is possibly too mild to develop vascular complications. In this regard, Wu et al. showed that 5 months of PM supplementation limited aortic stiffening and vascular resistance through reducing collagen glycation in 20-month-old mice [45].

### 3.3. Safety and Efficacy of PM

In our study, we administered PM with a dose of 1 g/L, which is commonly applied in animal research [17,30,35]. In the clinic, PM as a drug, with a dosage ranging from 100 mg/day to 600 mg/day, has been demonstrated to be safe and effective in multiple pathologies. In a randomized phase 2 clinical trial, PM was shown to improve kidney function, indicated by decreased serum creatinine, in diabetic patients with kidney disease [46]. A safety analysis reported no adverse events as well as no detrimental effects on the patient’s electrocardiogram due to PM treatment. Although Lewis et al. failed to detect the efficacy of PM on severe kidney dysfunction in diabetic patients, they suggested that patients with less advanced renal damage may have benefited from the drug [47]. Furthermore, PM supplementation was found to be tolerable and to decrease pain symptoms and systemic inflammation in patients with osteoarthritis [48]. Recently, a randomized double-blind placebo-controlled trial was conducted to investigate the effect of PM on glycation as well as metabolic and vascular function in relatively healthy but abdominally obese individuals [49]. Despite having no effect on weight, insulin sensitivity, microvascular function, and blood pressure, which aligns with our findings, PM was potent enough to improve circulating glycation (e.g., methylglyoxal) and endothelial dysfunction markers in the study population. Whether PM will be effective in improving adverse cardiac remodeling and function in (pre)diabetic individuals remains to be investigated.

### 3.4. Limitations

The outbred nature of the Sprague Dawley strain implies a varied susceptibility to hypercaloric diets in the development of T2DM with cardiac dysfunction [50]. This further clarifies the large within-group variation but also explains the difference in observed DCM phenotypes among different in vivo studies [25,26]. Another limitation of this study is the lack of power, which could explain an absence of statistical significance in distinct echocardiographic and hemodynamic measurements. Despite the relatively low number included, clear differences and trends could be identified. Finally, the T2DM phenotype of our rat model is relatively mild. However, it displays a few hallmarks of developing adverse cardiac remodeling. Consequently, the PM dosage in the current study (1 g/L) may be potentially too low to identify its effects on cardiac pathophysiological markers.

## 4. Materials and Methods

### 4.1. Animal Model and Study Design

All animal experiments were performed in accordance with the EU directive 2010/63/EU for animal testing and were approved by the Local Ethical Committee for Animal Experimentation (UHasselt, Diepenbeek, Belgium; ID202073). All animals were housed in a temperature-controlled environment (21 °C, 60% humidity) on a 12 h light/dark cycle. Drinking water and food were available ad libitum. In total, 19 male Sprague Dawley rats (Janvier Labs, Le Genest-Saint-Isle, France) were used in this study. Healthy rats, at 6 weeks of age and weighing 250–270 g, were randomly divided into three groups (Figure 6). The first group received a CD (23% kcal% proteins, 13% kcal% fat, 64% kcal% carbohydrates from grains, no added sugars; Ssniff Spezialdiäten, VRF-1, Soest, Germany) (n = 7) for 18 weeks. The second group was fed a high-sugar or so-called WD (15% kcal proteins, 16% kcal fat, 69% kcal carbohydrates, of which were 48% kcal sugars from sweetened condensed milk and added sucrose) (n = 4) for 18 weeks to induce prediabetes with adverse cardiac remodeling and dysfunction, as described previously [25,26,51]. The third group was fed the WD and drinking water supplemented with PM dihydrochloride (1 g/L, sc-219673, Santa Cruz Biotechnology, Inc., Heidelberg, Germany) (n = 8) throughout the entire study [15,35].

Body weight was measured weekly. An oral glucose tolerance test (OGTT) was conducted at baseline, week 6, week 12, and week 18. LV echocardiographic measurements and blood sampling were performed at baseline and week 18. Invasive LV hemodynamic measurements were conducted at the end of the protocol, just before sacrifice. All procedures were performed under 2% isoflurane anesthesia supplemented with oxygen. At sacrifice, the animals were injected with heparin (1000 u/kg, intraperitoneal (i.p.)) to avoid blood clotting and euthanized by injection with an overdose of sodium pentobarbital (200 mg/kg, i.p., Dolethal, Vetoquinol, Aartselaar, Belgium). The heart, liver, and lungs were harvested and weighed. Transversal sections of the heart were fixed in 4% paraformaldehyde overnight, transferred to 70% ethanol, and embedded in paraffin for histological staining. The remaining LV tissue was snap-frozen in liquid nitrogen, crushed into a fine powder, and stored at −80° for analysis with Real-Time quantitative polymerase chain reaction (RT-qPCR). A hind leg per rat was removed for the measurements of tibia bone length to normalize organ weights.

### 4.2. Oral Glucose Tolerance Test and Insulin Measurements

Glucose tolerance was assessed at baseline, week 6, week 12, and week 18 after the onset of the diet, with a 1 h OGTT. After overnight fasting, glucose (2 g/kg) was administered through oral gavage. Blood glucose concentration was determined from capillary tail blood collection with Analox GM7 (Analis SA, Namur, Belgium) before glucose administration and repeated 15, 30, and 60 min after administration. Glucose response was expressed as the total area under the curve (AUC). Plasma insulin concentrations were measured using electrochemiluminescence (Meso Scale, Gaithersburg, MD, USA) at baseline and after 60′. The HOMA-IR was used to assess insulin resistance [52]. HOMA-IR was calculated from fasting glucose and insulin values through the use of the following formula: HOMA-IR = (fasting insulin [µIU/mL] × fasting glucose [mmol/L])/22.5.

### 4.3. Conventional Echocardiographic Measurements

The transthoracic echocardiography of the LV was conducted in a supine position under 2% isoflurane anesthesia supplemented with oxygen in all rats with a Vevo^®^ 3100 high-resolution imaging system and a 21 MHz MX250 transducer (FUJIFILM VisualSonics, Inc., Amsterdam, The Netherlands), as described previously [26,51]. While scanning, physiological parameters including heart rate (HR) and electrocardiogram signals were monitored noninvasively while measurements were taken. From parasternal short-axis views in M-mode, PWT_d_, PWT_s_, AWT_d_, AWT_s_, IVS_d_, IVS_s_, ID_d_, ID_s_, and radial FS were determined, for which the measurements of three heartbeats were averaged. ESV, end-diastolic volume (EDV), SV, cardiac output (CO), diastolic area (A_d_), A_s_, EF, and longitudinal FS were analyzed from parasternal long-axis views in B-mode. EDV, ESV, A_d_, A_s_, SV, and CO were normalized to body surface area (BSA), and, through this, the latter two parameters were expressed as the SV index and cardiac index. Apical four-chamber views in B-mode were used to receive mitral flow profiles. Pulsed wave Doppler mode was used to assess E/A. Peak septal mitral annulus velocity in the early filling phase (E’) was assessed with tissue Doppler. For pulsed wave and tissue Doppler, measurements from two cycles were averaged. The E/E’ ratio was calculated. Echocardiographic images were analyzed using the Vevo^®^ LAB software (Vevo^®^ LAB software, version 5.6.1, FUJIFILM VisualSonics, Inc.). Two researchers analyzed the echocardiographic images blinded.

### 4.4. Hemodynamic Measurements

Invasive LV pressure measurements were conducted at sacrifice, as described elsewhere [26,51]. Hemodynamic parameters were measured with a pre-calibrated SPR-320 MikroTip high-fidelity pressure catheter (Millar Inc., The Hague, The Netherlands) inserted via the right carotid artery into the LV. The pressure catheter was connected to a quad-bridge amplifier and PowerLab 26T module (AD Instruments, Oxford, UK) to transfer the data to LabChart v7.3.7 software (AD Instruments) for analysis. Hemodynamic parameters, including EDP, ESP, and Tau, were analyzed.

### 4.5. Histology

Transverse sections 7 µm thick were obtained from paraffin-embedded LV tissue at the midventricular level and aortic tissue. LV sections were stained with H&E to assess the CSA through the use of Fiji v1.53c software [53]. For this, an elliptical cross-section of a cardiomyocyte was assumed and at least 12 cells per animal were analyzed.

In addition, the LV and aortic sections were stained with the Sirius Red/Fast Green Collagen Staining kit (9046, Chondrex Inc., Woodville, TX, USA) following the manufacturer’s guidelines. After staining, sections were dehydrated in elevated concentrations of ethanol and mounted with a DPX-mounting medium. Images were captured with the Leica MC170 camera connected to a Leica DM2000 LED microscope (Leica Microsystems, Diegem, Belgium). The area of collagen deposition, indicated by red staining, was quantified in the LV and aorta in four randomly chosen and blinded images per section using the color deconvolution plugin in Fiji v1.53c software [53]. The area of collagen deposition was normalized to the total cardiac or aortic area and expressed as a percentage of collagen deposition. Histological analyses were performed blinded by two operators independently.

### 4.6. Immunohistochemistry

Transverse sections 7 µm thick were obtained from paraffin-embedded LV tissue at the midventricular level and aortic tissue. The LV sections were stained for AGEs, 3-nitrotyrosine, and 4-HNE, and the aortic sections were stained for 3-nitrotyrosine. Deparaffinized tissue sections underwent heat-mediated antigen retrieval with citrate buffer (pH = 6). Endogenous peroxidase was blocked with 30% hydrogen peroxide diluted 1:100 in 1X PBS for 20 min at room temperature (RT). The sections were permeabilized with 0.05% Triton X-100 (Merck Life Science BV, Overijse, Belgium) for 20 min at RT, and blocked with serum-free protein block (X0909, Dako, Agilent Technologies, Diegem, Belgium) for 20 min at RT to limit background staining. Consequently, the LV sections were incubated with a primary antibody against 3-nitrotyrosine (1:100, 1 h at RT, mouse monoclonal, ab7048, Abcam, Cambridge, UK), 4-HNE (1:400, overnight at 4°, mouse monoclonal, ab48506, Abcam, Cambridge, UK) or AGEs (1:250, 1 h at RT, rabbit polyclonal ab23722, Abcam, Cambridge, UK), whereas the aortic sections were incubated with a primary antibody against 3-nitrotyrosine (1:100, 30 min at RT, mouse monoclonal, Ab7048, Abcam, Cambridge, UK) diluted in 1X PBS. For 3-nitrotyrosine staining, a biotinylated secondary antibody (1:100, 30 min at RT, rabbit anti-mouse, E0413, Dako) and streptavidin-horseradish peroxidase (1:400, 30 min at RT, P0397, Dako) were applied. For AGEs and 4-HNE staining, EnVision™ with Dual Link System horseradish peroxidase (K5007, 30 min at RT, anti-rabbit/anti-mouse, Dako) was used. A negative control, incubated without a primary antibody, was included in each staining. All sections were incubated with 3,3′-diaminobenzidine (Dako) and were counterstained with hematoxylin to stain nuclei. The sections were dehydrated with increasing concentrations of ethanol and mounted using a DPX-mounting medium. Images were acquired using a Leica MC170 camera connected to a Leica DM2000 LED microscope (Leica Microsystems, Diegem, Belgium). The level of staining was assessed in four randomly chosen and blinded fields per section using the color deconvolution plugin in Fiji v1.53c software and was expressed as the percentage of the total cardiac or aortic area [53]. Two researchers blinded for group allocation performed the analysis independently.

### 4.7. RT-qPCR

Total RNA was extracted from 30 mg of snap-frozen LV tissue using the RNeasy Fibrous Tissue Kit (Qiagen Benelux B. V., Antwerp, Belgium) by following the manufacturer’s instructions. The concentration and purity of the RNA were assessed using the NanoDrop 2000 spectrophotometer (Isogen Life Science B. V., Utrecht, The Netherlands). cDNA synthesis was performed using qScript cDNA SuperMix (QuantaBio, VWR International bvba, Leuven, Belgium). Primers were designed in the coding sequence of the mRNA (Integrated DNA Technologies, Leuven, Belgium) (Appendix A). The expression of collagen type I, collagen type III, GPx1, NOX4, RAGE, and SOD2 was studied using a MicroAmp™ Fast Optical 96-well reaction plate with SYBR Green (Thermofisher Scientific, Geel, Belgium) and using the QuantStudio 3 PCR system (Thermofisher Scientific). Gene expression data were analyzed via the ∆∆CT method in accordance with the MIQE guidelines (58). The most stable reference genes for this experiment, namely phosphoglycerate kinase 1 (PGK1) and hypoxanthine-guanine phosphoribosyl transferase (HPRT), were averaged with geNorm for normalization of target gene expression.

### 4.8. Plasma Advanced Glycation End Products Determination

Plasma AGEs were determined with an OxiSelect Advanced Glycation End Product Competitive ELISA kit, according to the manufacturer’s instructions (Bio-connect, Huissen, The Netherlands) [25,26]. In brief, the plate was coated with AGEs conjugate overnight. Samples were added to the AGEs conjugate-precoated plate. Subsequently, wells were incubated with the primary anti-AGEs polyclonal antibody for 1 h, washed with wash buffer, and incubated with a horseradish peroxidase-conjugated secondary antibody for 1 h. Substrate solution was added to the wells for 10 min and absorbance was measured at 450 nm.

### 4.9. Statistical Analysis

Statistical analyses were conducted using GraphPad Prism (GraphPad Software, version 9.3.0, San Diego, CA, USA). Outliers were detected using the ROUT test method with a maximum desired False Discovery Rate of 1%. The normal distribution of data was evaluated with the Shapiro–Wilk test. Data that were normally distributed were subjected to the parametric one-way ANOVA test followed by Tukey’s multiple comparisons test. If the data were not normally distributed, a non-parametric Kruskal–Wallis test followed by Dunn’s multiple comparison test was applied. Data measured at multiple time points were compared with a repeated measures two-way ANOVA test followed by the Bonferroni post hoc correction. All data are expressed as mean ± standard error of the mean (SEM). Sample size is indicated as ‘n’. *p* < 0.05 was considered statistically significant.

## 5. Conclusions

In this study, we uncovered that PM intervention decreases cardiac interstitial fibrosis and oxidative stress in WD-fed rats. As PM could lower these disease markers, PM potentially offers promise for alleviating worse cardiac outcomes in patients with T2DM or those at risk for T2DM.

## Figures and Tables

**Figure 1 ijms-25-08508-f001:**
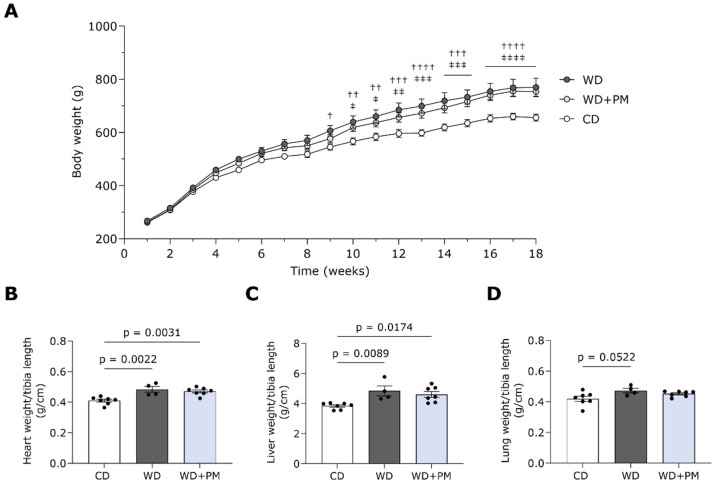
Pyridoxamine does not affect animal characteristics of Western diet-fed rats. Biometric characteristics of all groups including CD (n = 7), WD (n = 4), and WD + PM (n = 7). (**A**) Progression of body weight over time. (**B**) Heart weight to tibia length ratio. (**C**) Liver weight to tibia length ratio. (**D**) Wet lung weight to tibia length. Data represent mean ± SEM. † denotes *p* < 0.05, †† denotes *p* < 0.01, ††† denotes *p* < 0.001, and *p* < 0.0001 denotes †††† CD vs. WD. ‡ denotes *p* < 0.05, ‡‡ denotes *p* < 0.01, ‡‡‡ denotes *p* < 0.001, and *p* < 0.0001 denotes ‡‡‡‡ CD vs. WD + PM.

**Figure 2 ijms-25-08508-f002:**
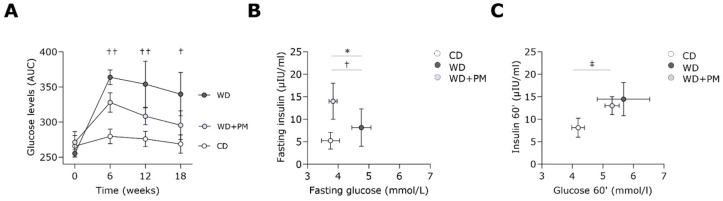
Pyridoxamine lowers fasting blood glucose levels in Western diet-fed rats. (**A**) Glucose levels, expressed as AUC over time, were measured during a 1 h OGTT in plasma from CD (n = 7), WD (n = 4), and WD + PM (n = 8). Insulin levels as a function of glucose levels were obtained at (**B**) fasting or (**C**) 60′ post-glucose administration during an OGTT in plasma from CD (n = 6), WD (n = 4), and WD + PM (n = 7). Data represent mean ± SEM. † denotes *p* < 0.05 and †† denotes *p* < 0.01 CD vs. WD. ‡ denotes *p* < 0.05 CD vs. WD + PM. * denotes *p* < 0.05 WD vs. WD + PM. AUC, area under the curve. OGTT, oral glucose tolerance test.

**Figure 3 ijms-25-08508-f003:**
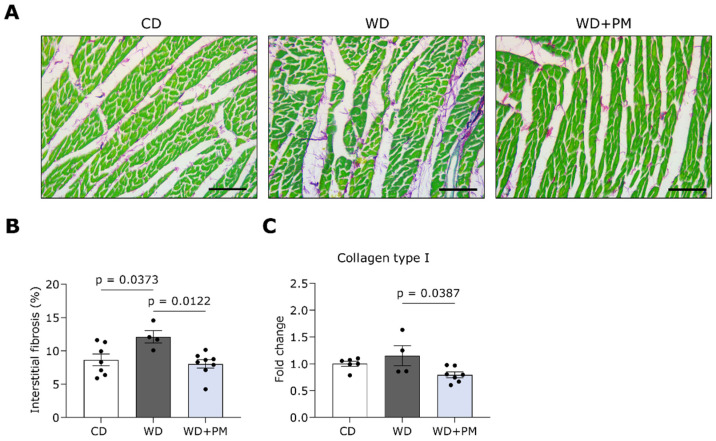
Pyridoxamine prevents LV collagen deposition in Western diet-fed rats. (**A**) Representative pictures of LV tissue stained with Sirius Red/Fast Green. Fibrotic tissue is stained red/purple while cardiac cells are stained green. The scalebar represents 100 µm. (**B**) Quantification of the percentage of interstitial collagen deposition per surface area in LV tissue from CD (n = 7), WD (n = 4), and WD + PM (n = 8). (**C**) Quantification of gene expression of collagen type I in LV tissue from CD (n = 6), WD (n = 4), and WD + PM (n = 8). Data represent mean ± SEM. LV, left ventricular.

**Figure 4 ijms-25-08508-f004:**
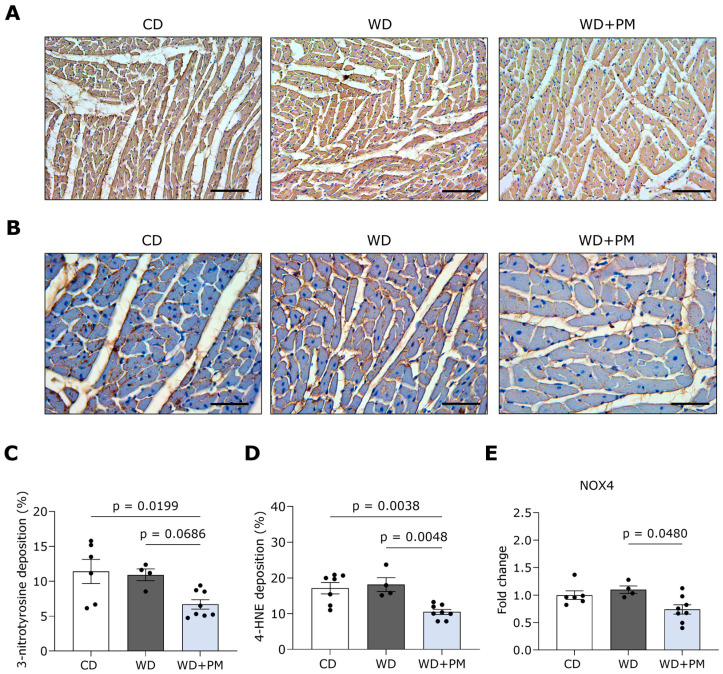
Pyridoxamine tends to limit LV oxidative stress in Western diet-fed rats. (**A**) Representative pictures of LV tissue stained for 3-nitrotyrosine (brown). The scalebar represents 100 µm. (**B**) Representative pictures of LV tissue stained for 4-HNE (brown). The scalebar represents 50 µm. (**C**) Quantification of 3-nitrotyrosine deposition per surface area in LV tissue from CD (n = 7), WD (n = 4), and WD + PM (n = 8). (**D**) Quantification of 4-HNE deposition per surface area in LV tissue from CD (n = 7), WD (n = 4), and WD + PM (n = 8). (**E**) Quantification of gene expression of NOX4 in LV tissue from CD (n = 6), WD (n = 4), and WD + PM (n = 8). Data represent mean ± SEM. 4-HNE, 4-hydroxynonenal. LV, left ventricular. NOX4, nicotinamide adenine dinucleotide phosphate oxidase 4.

**Figure 5 ijms-25-08508-f005:**
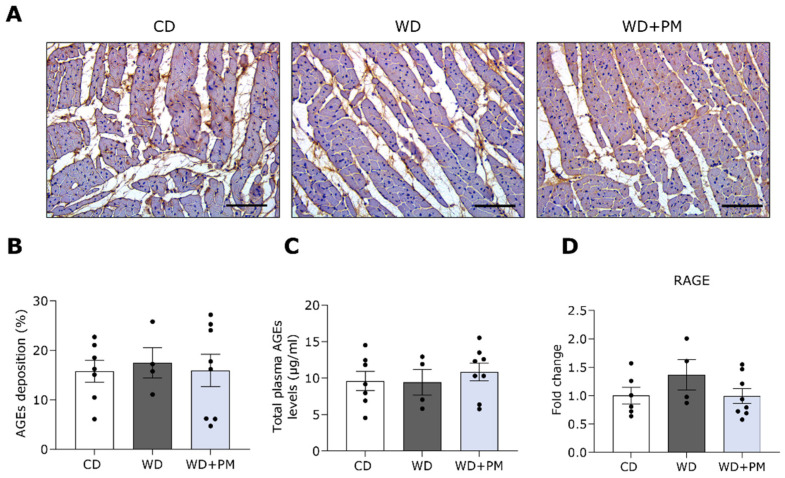
Pyridoxamine does not affect the total advanced glycation end product content in the LV and plasma of Western diet-fed rats. (**A**) Representative pictures of LV tissue stained for total AGEs deposition (brown). The scalebar represents 100 µm. (**B**) Quantification of AGEs deposition in LV tissue from CD (n = 7), WD (n = 4), and WD + PM (n = 8). (**C**) Quantification of total plasma AGEs from CD (n = 6), WD (n = 4), and WD + PM (n = 8). (**D**) Quantification of gene expression of RAGE in LV tissue from CD (n = 6), WD (n = 4), and WD + PM (n = 8). Data represent mean ± SEM. AGEs, advanced glycation end products. LV, left ventricular. RAGE, receptor for AGEs.

**Figure 6 ijms-25-08508-f006:**
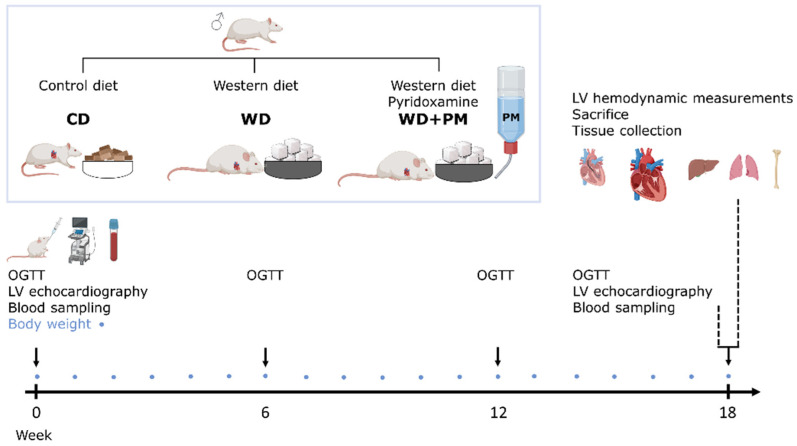
Experimental design of the study. Male Sprague Dawley rats received a standard chow diet (n = 7), Western diet (n = 4), or Western diet and drinking water supplemented with PM (n = 8) for 18 weeks in parallel. Body weight was measured weekly, indicated by blue circles. An OGTT was performed at baseline, week 6, week 12, and week 18. LV echocardiography together with blood sampling was performed at baseline and week 18. Invasive hemodynamic measurements of the LV were performed at sacrifice. The heart, liver, lungs, and tibia were isolated. LV, left ventricle. CD, standard chow diet. OGTT, oral glucose tolerance test. WD, Western diet. PM, pyridoxamine.

**Table 1 ijms-25-08508-t001:** LV echocardiographic parameters at week 18.

	CD	WD	WD + PM
EDV/BSA (µL/cm^2^)	0.841 ± 0.046	1.037 ± 0.104	0.866 ± 0.043
ESV/BSA (µL/cm^2^)	0.196 ± 0.027	0.327 ± 0.028 *	0.232 ± 0.028
A_d_/BSA (mm^2^/cm^2^)	0.149 ± 0.007	0.157 ± 0.009	0.144 ± 0.005
A_s_/BSA (mm^2^/cm^2^)	0.062 ± 0.005	0.081 ± 0.002 *	0.063 ± 0.005
EF (%)	78 ± 2	68 ± 3	73 ± 3
HR (bpm)	326 ± 15	340 ± 10	341 ± 10
BSA (cm^2^)	739 ± 10	822 ± 24 *	811 ± 14 **

Echocardiographic characteristics at the end of the study of CD (n = 7), WD (n = 4), and WD + PM (n = 8). Data represent mean ± SEM. * denotes *p* < 0.05 and ** denotes *p* < 0.01 vs. CD. A_s_: *p* = 0.0886 WD vs. WD + PM. EF: *p* = 0.0850 CD vs. WD. ESV: *p* = 0.1058 WD vs. WD + PM. A_d_, diastolic area. A_s_, systolic area. BSA, body surface area. EDV, end-diastolic volume. EF, ejection fraction. ESV, end-systolic volume. HR, heart rate.

## Data Availability

The data presented in this study are available on request from the corresponding author.

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
