# Peer review of "Pyridoxamine Alleviates Cardiac Fibrosis and Oxidative Stress in Western Diet-Induced Prediabetic Rats"

_ijms, 2024, doi:10.3390/ijms25158508_

Round 1

Reviewer 1 Report (Previous Reviewer 2)

Comments and Suggestions for Authors

It has been improved according to comments. Hope this publication will be helpful in human health promotion.

Author Response

Reviewer 2 Report (New Reviewer)

Comments and Suggestions for Authors

The manuscript, titled Pyridoxamine alleviates cardiac fibrosis and Oxidative Stress in Western diet-induced prediabetic rats, is interesting and novel.

The authors tried to show the protective role of Pyridoxamine (PM) in Western diet-induced prediabetes and adverse cardiac remodeling and dysfunction. They employed the rat model to study the effects of PM in vivo. PM administration has been shown to reduce blood glucose levels in WD-induced T2DM rats. Additionally, PM attenuated LV dilation in WD-feed rats. Although the findings of this study relied on in vivo animal models, including some in vitro cell culture study would benefit the authors to concrete their findings.

1.     Elevated 4HNE (4-hydroxy-2-nonenal) adduct formation is an important marker of T2DM-induced oxidative stress and has also been implicated in cardiac dysfunction. Testing the 4HNE protein adduct levels in the LV tissues from all the experimental groups would benefit the authors in finding the molecular targets of WD-induced cardiac fibrosis.

 2.     It is already known that WD-fed mice and rats develop cardiac hypertrophy. Therefore, the authors can measure the cardiomyocyte cross-sectional area, length, and width to study the role of PM in rescuing the phenotype of WD-induced cardiac hypertrophy.

 3.     I suggest the authors include another experimental group, Chow diet (CD) + PM, to see the effects of PM on the controls.

Round 2

Reviewer 2 Report (New Reviewer)

Comments and Suggestions for Authors

I want to thank the authors for addressing all of my concerns with the previous version of the manuscript! Therefore, I strongly recommend publishing the revised version of the manuscript. 

This manuscript is a resubmission of an earlier submission. The following is a list of the peer review reports and author responses from that submission.

Round 1

Reviewer 1 Report

Comments and Suggestions for Authors

D’Haese et al. investigated the effects of pyridoxamine on heart morphology, function, and molecular markers of nitro-oxidative stress in rats with prediabetes induced by Western diet. Their major finding is that pyridoxamine reduced cardiac fibrosis compared to the WD group and oxidative stress compared to the control group. The authors used different methods, and the text is easy to read. 

1. The sample size (n=4) in the DM group is low. Why did the authors not plan a higher sample number (n=8) in this group, similar to the other two groups? In the present form, this seems to be a preliminary study.

2. Was there any difference in the water consumption of the groups? Why did the authors not use per os (oral gavage) administration of pyridoxamine to ensure the same dose in the treated animals? There is a similar concern with the high-sugar diet. How can you ensure that animals get the same dose (g sucrose/kg) or higher calories in the WD groups?

3. The echocardiography and histology results contradict each other. How could the WD group not develop diastolic dysfunction despite the significant fibrosis, LV dilation, and compensatory hypertrophy? Normally, diastolic dysfunction precedes systolic dysfunction. Please also show the absolute values of the heart, LV, and other organ weights. 

4. The authors claim they measured the wall thicknesses in the short-axis view and the LV diameters with other parameters (e.g., EF) in the long-axis view. Please also show the septal wall thicknesses and the LV diameters from a short-axis view. Beyond the EF, the FS could be informative on systolic function. Please show the FS also.

5. It would be helpful to measure the cardiac expression changes of hypertrophy (Myh6, Myh7), remodeling, and fibrosis (e.g., Mmp2, Mmp9, Col1a1, Tgfb, Smad2, and Smad3, etc.), heart failure and cardiac stretch markers (e.g., Nppa, Nppb, etc.).

6. Vitamin B6 is a cofactor of aminotransferases (e.g., ALAT, ASAT) that can be expressed in the heart and/or liver and are also associated with cardiovascular risk. The plasma and tissue levels could also be interesting in this study. 

7. The nitrotyrosine and AGE levels and the Nox4 expression were similar in the control and WD groups. Please discuss why these markers failed to increase in your prediabetes model. What induced the cardiac fibrosis in your WD model, if not the oxidative stress and AGEs? Please explain it in the Discussion.

8. Pyridoxamine decreased the nitro-oxidative stress compared to the control group. The title, abstract, and discussion are misleading in the present form. Please rephrase them accordingly.

Reviewer 2 Report

Comments and Suggestions for Authors

Pyridoxamine (PM), a vitamin B6 analog, has been shown to exert protective effects in metabolic and cardiovascular diseases. Current report used the rats received a standard chow diet or Western diet (WD) for 18 weeks as animal model of prediabetes to investigate the effect of PM. I like to give the following comments.

1.      Diabetic patients with systolic dysfunction or heart failure with reduced ejection fraction (HFrEF) is similar to this animal model that needs to introduce in clear.

2.      In the introduction section, PM offers merits in diet-induced prediabetic rats that needs reference(s) to follow.

3.      Current report is going to show that PM prevents WD-induced interstitial fibrosis and oxidative stress in the heart. However, positive reference was not used in parallel.

4.      Data in Table 1 lacked the significant change, especially the ejection fraction (EF). Why?

5.      It has been described that PM at 1 g/L for 9 weeks may improve the hyperglycemia but did not influence insulin levels or body weight in mice fed a moderate-fructose and -fat diet [30]. How to link with current report?

6.      In the limitations, measurement of cardiac contractility in Langendroff apparatus may provide the reliable data.

7.      Drinking water supplemented with PM dihydrochloride at 1 g/L that needs reference(s) to support. Plasma AGEs assay also needs the reference(s).

8.      Conclusions seem too weak. Please rephrase it.

Comments on the Quality of English Language

It seems better to check through a professional editing service.
